# Self-Supervised Convolutional Neural Network Learning in a Hybrid Approach Framework to Estimate Chlorophyll and Nitrogen Content of Maize from Hyperspectral Images

Ignazio Gallo [1,*,†], Mirco Boschetti [2,†], Anwar Ur Rehman [1,†] and Gabriele Candiani [2,†]

1   Department of Theoretical and Applied Science, University of Insubria, 21100 Varese, Italy;
    aurehman1@uninsubria.it
2   Institute for Remote Sensing of Environment, Consiglio Nazionale delle Ricerche, 20133 Milano, Italy;
    boschetti.m@irea.cnr.it (M.B.); candiani.g@irea.cnr.it (G.C.)
*   Correspondence: ignazio.gallo@uninsubria.it
†   These authors contributed equally to this work.

**Abstract:** The new generation of available (i.e., PRISMA, ENMAP, DESIS) and future (i.e., ESA-CHIME, NASA-SBG) spaceborne hyperspectral missions provide unprecedented data for environmental and agricultural monitoring, such as crop trait assessment. This paper focuses on retrieving two crop traits, specifically Chlorophyll and Nitrogen content at the canopy level (CCC and CNC), starting from hyperspectral images acquired during the CHIME-RCS project, exploiting a self-supervised learning (SSL) technique. SSL is a machine learning paradigm that leverages unlabeled data to generate valuable representations for downstream tasks, bridging the gap between unsupervised and supervised learning. The proposed method comprises pre-training and fine-tuning procedures: in the first stage, a de-noising Convolutional Autoencoder is trained using pairs of noisy and clean CHIME-like images; the pre-trained Encoder network is utilized as-is or fine-tuned in the second stage. The paper demonstrates the applicability of this technique in hybrid approach methods that combine Radiative Transfer Modelling (RTM) and Machine Learning Regression Algorithm (MLRA) to set up a retrieval schema able to estimate crop traits from new generation space-born hyperspectral data. The results showcase excellent prediction accuracy for estimating CCC ($R2 = 0.8318$; RMSE = 0.2490) and CNC ($R2 = 0.9186$; RMSE = 0.7908) for maize crops from CHIME-like images without requiring further ground data calibration.

**Keywords:** self-supervised learning; hybrid approach; deep learning; convolutional neural network; hyper-spectral images

## 1. Introduction

Maize is the third most widely produced crop worldwide, with numerous social and economic benefits in food, livestock, bio-fuels, industrial uses, and employment [1]. In recent years, due to global warming, bad weather has been routinely occurring worldwide, significantly affecting agricultural production, reducing yield, and increasing the possibility of impacts such as crop lodging. In this context, Nitrogen (N) represents an important macro-nutrient for maize crops playing a vital role in its growth. Inadequate N can hurt photosynthetic efficiency and chlorophyll (Cab) production, an important factor for maize growth [2]. Therefore, precisely monitoring N and Cab content in maize crops is important for quality growth.

The techniques for estimating N and Cab concentrations and making yield forecasts include destructive and Non-Destructive (ND) methods. Both of these have pros and cons for correctly estimating crop nutrients. The destructive method is reliable and precise. However, it mainly relies on tissue examination of maize leaves in the laboratory; data

collection over a vast area is expensive and takes a long time, making it unfeasible. Consequently, a quick, inexpensive, and environmentally acceptable method is needed for the scheduled nutritional status inspection of maize crops. The ND techniques facilitate the prompt and accurate estimation of crops' N and Cab levels without harming the crops [3]. The ND methods have been developed to derive accurate data on plant functional traits, including nutrient levels, using canopy/leaf reflectance [4]. These methods usually exploit visible and red-edge portions of the electromagnetic spectrum that are the more correlated with chlorophyll content [5,6].

With the recent advancement of remote sensing innovation, the optical sensors onboard satellites, airplanes, and Unmanned Aerial Vehicles (UAV) are used as input for an ND, rapid, and considerably less expensive crop nutrients estimation approach. These remote sensors remotely record the crop information in the form of Red Green Blue (RGB), spectral, multi-spectral, and Hyper-Spectral (HS) data, which represent the radiation reflected by plants [7]. Optical remote sensors impart huge amounts of spectral data in several regions covering visible, near-infrared, and shortwave ranges. The spectral data of these regions are used indirectly to evaluate the morphological (canopy density, leaf area) and biochemical and physiological properties (chlorophyll and photosynthesis pigments) of the crops and consequently derive the Cab and N content [8]. Typically, these remote sensors deliver data in a cubic format, including spatial data in two dimensions, X and Y, and spectral data in the Z axis. We can split the estimation techniques into two groups based on the dimensionality of the input data: spatial analysis and spectral analysis.

The spectral analysis techniques employed in remote sensing applications assume that the spectral data of each pixel can be utilized to infer the presence and quantify specific target properties that in the crop are related to its traits, such as Cab and N content [9]. The most general spectral analysis technique uses vegetation indexes (VI) based on particular wavelengths. However, determining crops' nutrient (N and Cab) content indirectly from remotely sensed information is considered a challenging issue influenced by several important factors, including variations in crop varieties and different environments. The VI techniques based on certain wavelengths are considered sensitive to these factors, and non-generalizable [10]. Machine Learning (ML) has demonstrated its efficacy in resolving nonlinear issues from various sources [11] and, in recent years, crop N estimation has become increasingly popular. In [12,13], the authors applied three ML models—Artificial Neural Networks (ANN), Random Forest (RF), and Support Vector Machine (SVM)—to estimate the N content of rice crops using all of the available spectral data. The RF used in these two papers showed the highest accuracy and strongest generalization performance.

Deep learning (DL) is a subset of ML that focuses on artificial neural networks with multiple layers, enabling the model to automatically learn hierarchical representations of data and extract intricate patterns and features, leading to more sophisticated and abstract decision-making capabilities. However, in the context of ML, directly applying DL models to estimate crops' N and Cab content still deteriorates from the following issues. First, most existing DL models' structures are made to capture spatial information, and these models lack a dedicated module for learning spectral information, which is crucial for estimating crop N status. Second, DL techniques are naturally data-hungry and require big datasets (labeled data) for model training to achieve good performance and prevent over-fitting. Finally, DL models have very high computational complexity, making them difficult to scale well using a large volume of remote sensing data [3].

A data-driven approach for crop traits estimation suffers from limited exportability capacity in space (other sites), time (other seasons), and context (different crops/vegetation typology) concerning the dataset used for model calibration. For this reason, the state-of-the-art of crop trait retrieval from remote sensing is based on the so-called hybrid approach. This innovative methodology, combining physically-based Radiative Transfer Models (RTMs) with the flexibility and computational efficiency of Machine Learning Regression Algorithms (MLRAs), has been reported as the most promising solution [14]. RTMs can simulate the spectral response of plant canopies under a wide range of crop

conditions. The generation of simulated hyperspectral spectra is based on RTM equations that consider the effect of different leaf and canopy properties, background (i.e., reflectance) spectral condition, and considering observer and viewing geometry. In contrast, MLRAs can solve complex non-linear problems using artificial intelligence tools to analyze big data and identify underlying patterns. In this context, the simulation from RTM can provide the input data cardinality for the exploitation of DL models, exploiting extracted spectral features and crop traits parameters used in the simulation. Once the MLRA is trained on simulated data, it can be applied to real hyperspectral imagery. Generated trait maps are then validated to assess the robustness of MLRA with an independent dataset of field-measured samples.

SSL is a Transfer Learning (TL) technique that involves training a model on unlabeled data to learn useful representations, which can then be applied to downstream tasks with limited labeled data. In this study, we trained a self-supervised DL approach using real spectra to enhance the hybrid method. We learned the regression problem of maize crop traits such as Cab and N content at the canopy level (Canopy Chlorophyll Content (CCC) and Canopy Nitrogen Content (CNC)) from simulated crop data generated by PROSPECT-PRO [15]. The proposed method uses the information available in imagery combined with a novel SSL scheme that exploits the inherent correlations between crop traits and spectral features. As reported in Figure 1, this study presents two-fold learning methods. We used clean and noisy HS images in the first fold and trained a de-noising CNN. In the second fold, the pre-trained CNN part is used for feature extraction and integrated into an MLP to capture the spectral correlation in the data provided by the PROSAIL-PRO simulation. The hypothesis is that the pre-trained model has a good representation network as it transfers the knowledge of the previously trained model to the new small dataset and removes the need for a more extensive dataset.

**Figure 1.** Schematic representation of the training process of the proposed approach. The CNN autoencoder is trained using an unsupervised dataset (panel **A**). In contrast, the final model composed of a pre-trained encoder and an MLP is trained on simulated data (panel **A**) and used for regression on real HS images (panel **B**).

The main objectives of this study can be summarized as follows:

- SSL is a machine learning paradigm that utilizes unlabeled data to learn valuable representations and supervisory signals, in our case, without human-annotated labels. The objective is to investigate how SSL methods can process unlabeled hyperspectral data, effectively capturing spectral correlations and exploiting them on simulated data to learn how to retrieve crop traits.
- For this purpose, the study proposes an innovative two-step SSL learning method consisting of pre-training and further training procedures. In the first stage, a convolutional neural network (CNN) for de-noising is trained using pairs of noisy and clean images. In the second stage, the pre-trained network is utilized to identify the spectral correlation between latent features and crop traits.
- The effectiveness of the proposed two-stage learning method in estimating CCC and CNC in maize crops from hyperspectral images is evaluated. The objective is to demonstrate the predictive capabilities and accuracy of the proposed technique in estimating these crop traits using hyperspectral images. The performance is compared with other proposed results to better assess the obtained results.
- With our results, we demonstrate the effectiveness of the proposed approach in accurately estimating CCC and CNC and its potential for advancing the monitoring and management of maize crop productivity and ecosystem health based on satellite data.

## 2. Related Work

TL has been extensively explored within the computer vision community [16]. It involves using pre-trained models as a starting point for training new models on different tasks or datasets. TL has shown to be a highly effective technique that allows models to leverage knowledge gained from previously learned tasks to perform well on new, unseen tasks. In recent years, SSL has emerged as a promising approach to pre-train deep neural networks, which can then be fine-tuned for supervised downstream tasks. SSL techniques leverage the abundance of unlabeled data to learn valuable and generalizable representations. Instead of relying on human-labeled annotations, SSL manipulates the inherent structure and context within the data to learn meaningful representations. A common approach is to use pretext tasks, such as predicting the missing portion of an image or audio segment, as a proxy for learning meaningful representations that can then be transferred to downstream tasks.

Recent works have shown that SSL can achieve state-of-the-art results on various tasks and outperform supervised learning in specific contexts. For instance, Chen et al. [17] proposed SimCLR, a simple framework for Contrastive Learning (CL) that uses data augmentations to learn representations invariant to variations in viewpoint, scale, and color. Another notable example is the work by He et al. [18], who introduced momentum contrast (MoCo), a self-supervised approach that uses a memory bank and a momentum update strategy to improve the quality of the learned representations. Another study [19] investigated the potential of SSL using pre-trained weights for agricultural images. They used CL with Un-Supervised Learning (USL). They utilized the Grassland Europe, Areaila Farmland [20], and DeepWeeds [21] agricultural datasets and generated their pre-trained weights, then used these weights as a TL for unannotated agricultural images for the related task, like plant classification and segmentation. Their results outperform traditional DL approaches and achieved an up to 14% increase in average top-1 classification accuracy. The authors claimed that generating a domain-specific pre-trained weight is easy, helpful, and less computationally expensive. The work in [22] claimed that USL saves time, cost, and effort when labeling the data for supervised learning. The authors used different weather-based agricultural labeled data collected in 2018. They used its pre-trained weights to predict the new trends for the same type of unlabeled data and achieved higher accuracy.

SSL for HS images is an emerging technique that exploits the massive amount of unlabeled data available in these images. It leverages the inherent structure and statistics of the data to learn valuable representations without the need for direct supervision. Recent

studies have demonstrated its potential to improve different HS image analysis tasks, including classification, segmentation, and anomaly detection [23–25]. Various studies employed HS data and SSL to estimate the content of nutrients in different crops. The authors [3] presented a Self-Supervised Vision Transformer (SSVT) for correctly estimating $N$ in wheat crops using UAV data. The SSVT combines the spatial interaction block and spectral attention block to learn the spectral and spatial features from the data simultaneously. Further, the SSVT introduces local-to-global SSL to train the model using different unlabeled images data, and the proposed model outperforms by getting 96% accuracy compared to the original vision transformer, EfficientNetV2, EfficientNet, ResNet, and RegNet in training and testing. Another study [26] utilized the 576 HS data samples of maize crops by proposing a features enhancement approach of Competitive Adaptive Re-weighted Sampling and Long Short Term Memory (CARS-LSTM) hybrid model to improve the Cab content detection. In CARS-LSTM, CARS extracted the potential wavelength, and LSTM optimized the extracted features. This hybrid model demonstrated efficacy by obtaining the best Coefficient of Prediction set (RP2) of 94% and Root Mean Square Error of Prediction set (RMSEP) of 1.54 mg/L. Yin et al. [27] analyzed the effect of various altitudes while capturing the HS images. The experimental study used the fusion-based unsupervised classification to monitor the N content using 108 HS cotton crop data samples and reduced the error among the classified spectral samples. This work proved that using Gaussian processing with smoothing and Standard Normal Variate (SNV) can reduce the interference of redundancy in the HS images at 60 m and more than 60 m altitudes, respectively. Another study [28] implemented pixel-level multi-SSL for tree classification using HS data and multi-spectral different tree species images and extracted the features by combining the generative and contrastive learning approaches. The authors introduced multi-source adversarial and variational auto-encoders as a pretext step to capture multi-source features like data augmentation. Also, they used depth-wise cross-attention steps, which discriminate these features to get the effective ones. Their deep SSL study achieved 78% tree species accuracy using label-less HS validation data.

SSL has also been used for regression with HS histopathology images [29]. In this paper, the authors transform the HS images to a low-dimensional spectral embedding space using spectral regression, where the trained regression model is self-supervised. The approach does not require any labeled data, enabling the efficient use of available data. Candiani et al. [30] generated a synthetic version of a hyperspectral dataset using RTM modeling to create input for MLRA. The authors used Hybrid Approach (HYB) and Active Learning Heuristics (HAL) to manage the dimensionality reduction in HS data and evaluate the N and Cab crop trait retrieval at both canopy and leaf levels by performing mapping on two HS images simulated from an aerial hyperspectral acquisition. They concluded that the HYB approaches could successfully estimate the maize crops N and Cab from spectroscopy. These approaches can help researchers extract nutrients using other models from HS data. In the above literature, existing studies employed various HS images for the experiment. However, this study only utilized a single HS image for training and single HS image to test the estimation of maize crop traits using the CNN autoencoders SSL model.

## 3. Methodology

This work presents an SSL-based approach to estimate maize's N and Cab status from remote sensing Hyper-Spectral (HS) data. As shown in Figure 1, the proposed two-step methodology consists of (i) a convolutional-based autoencoders (encoder plus decoder) model pre-trained on HS data images to extract latent space features from HS data and (ii) a combination of the pre-trained encoder plus a Multi-Layer Perceptron (MLP) used for regression, to train a model for crop traits retrieval by exploiting RTM simulation. In the first stage of the methodology, to solve the data-hungry issue of DL models, SSL is used to pre-train the encoder with unlabeled data. In particular, the CNN autoencoder is trained to recreate the HS input signature and to remove artificial noise (added to the exploited HS

imagery) from the same input. In the second stage, the pre-trained encoder of the first stage is used, adding an MLP to regress maize crop traits. In a hybrid approach framework [14], this newly composed model is trained using synthetic data (PROSPECT-PRO simulation for maize). At the same time, generalization ability is measured against actual data (ground data from field campaign [30]). Further in this section, all the methodology modules are given in detail.

### 3.1. Autoencoder

Autoencoders [31] are a type of ANN that is designed to learn efficient features from unlabeled input data (unsupervised learning), which can further be used for data generation, denoising, anomaly detection, and many other things. Let us denote the hyperspectral input signatures as $X$ and the hyperspectral output signatures as $X'$. An encoder and a decoder are the two essential parts of an autoencoder.

The encoder maps the input data $X$ to a lower-dimensional representation, typically referred to as the "*latent space*" or "*encoding.*" It transforms the input data into a compressed representation that captures the most important features. Mathematically, the encoder can be denoted as $E(X) = Z$, where $E$ is the encoding function and $Z$ represents the encoded representation.

The decoder reconstructs the encoded representation $Z$ back to the original input space, attempting to generate an output $X'$ that closely resembles the input $X$. The decoder aims to replicate the input data from the compressed representation. The decoder can be mathematically denoted as $D(Z) = X'$, where $D$ is the decoding function.

The objective of the autoencoder is to minimize the difference between the input $X$ and the reconstructed output $X'$, quantified using the mean squared error (MSE) loss function in Equation (2).

$$L_{MSE} = (1/n) * \sum (X - X')^2 \tag{1}$$

A dataset comprising all the pixels $X$ of a hyper-spectral image is used as input data to train the autoencoder. To improve the generalization capabilities of the model, uniform random noise $N(\mu, \sigma)$ is added to input $X$ with a probability $p = 0.5$, where $\sigma \in [0.1, 0.0025]$ and $\mu = 0$. The new objective of the autoencoder is now to minimize the following loss function

$$L_{MSE} = (1/n) * \sum (X - D(E(X + N(\mu, \sigma))))^2 \tag{2}$$

Because we used only one single HS image to train the autoencoder, we used data augmentation to increase the variability inside the dataset. In particular, we created two augmentation techniques, GaussAmpl and RangeAmpl, which are added with a certain probability to the signal $X$. In this way, the neural network must also learn to regenerate $(X + \text{Augmentation})$ in output. As shown in the example of Figure 2, GaussAmpl adds a Gaussian function with random mean and standard deviation to the input signature. At the same time, the RangeAmpl selects a random range where a random constant is added to the signature.

The training process involves optimizing the encoder and decoder parameters to minimize the reconstruction loss using the backpropagation technique. The encoder transforms the input data into a compressed representation that captures the most essential features. Then, at the end of the autoencoder training process, we are interested in latent space extracted from the encoder by building on it a regressor to estimate the values of N and Cab we are interested in. The decoder is not used in estimating crop traits since its purpose was only parameter optimization through the backpropagation step. In Table 1, the configuration of the autoencoder used in this work is reported in the two sub-tables named "Encoder" and "Decoder." The meaning of their layers is explained in the following section. This approach can be seen as the action of dimensionality reduction reported as an efficient way to pre-process hyperspectral data before training a machine learning regression algorithm [14,30].

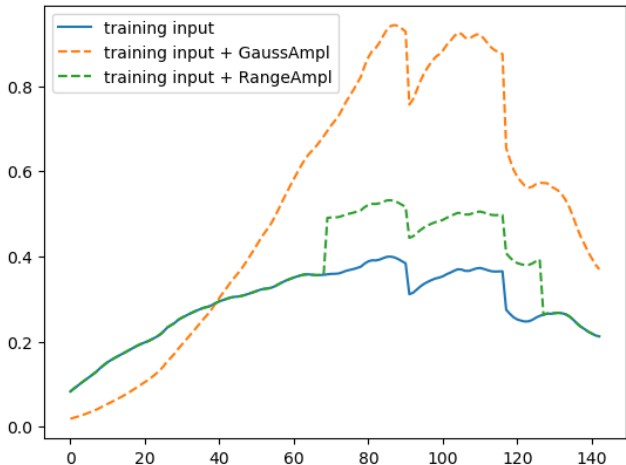

**Figure 2.** An example of an augmented hyper-spectral signature. The GaussAmpl adds a randomly created Gaussian function having mean = 103 and std = 47 to the input signature, while the RangeAmpl adds a random constant 0.13 to the signature in the random range = [69, 127].

**Table 1.** Autoencoder (Encoder plus Decoder) used during the experiments in this work. The *Conv1d (ni, no)* is the 1D convolution over an input signal, where *ni* and *no* is the number of input and output channels, respectively. The "Param" column reports the number of parameters for each layer. In the final regression model, the MLP is connected to the Encoder's output, and the Decoder will no longer be useful.

| Layer (Type) | Out. Shape | Param |
|---|---|---|
| Encoder | | |
| Conv1d (1, 24) | $[-1, 24, 143]$ | 96 |
| ReLU | $[-1, 24, 143]$ | 0 |
| Conv1d (24, 24) | $[-1, 24, 143]$ | 1752 |
| BatchNorm1d | $[-1, 24, 143]$ | 48 |
| ReLU | $[-1, 24, 143]$ | 0 |
| MaxPool1d | $[-1, 24, 71]$ | 0 |
| Conv1d (24, 12) | $[-1, 12, 71]$ | 876 |
| ReLU | $[-1, 12, 71]$ | 0 |
| Conv1d (12, 12) | $[-1, 12, 71]$ | 444 |
| BatchNorm1d | $[-1, 12, 71]$ | 24 |
| ReLU | $[-1, 12, 71]$ | 0 |
| MaxPool1d | $[-1, 12, 35]$ | 0 |
| Conv1d (12, 6) | $[-1, 6, 35]$ | 222 |
| ReLU | $[-1, 6, 35]$ | 0 |
| MaxPool1d | $[-1, 6, 17]$ | 0 |
| Decoder | | |
| Conv1d (6, 12) | $[-1, 12, 17]$ | 228 |
| ReLU | $[-1, 12, 17]$ | 0 |
| Interpolate | $[-1, 12, 57]$ | 0 |
| Conv1d(12, 24) | $[-1, 24, 57]$ | 888 |
| ReLU | $[-1, 24, 57]$ | 0 |
| Interpolate | $[-1, 24, 115]$ | 0 |
| Conv1d (24, 1) | $[-1, 1, 115]$ | 73 |
| ReLU | $[-1, 1, 115]$ | 0 |
| Interpolate | $[-1, 1, 143]$ | 0 |

**Table 1.** *Cont.*

| Layer (Type) | Out. Shape | Param |
|:---:|:---:|:---:|
| MLP for regression | | |
| Flatten | $[-1, 102]$ | 0 |
| Linear (102, 51) | $[-1, 51]$ | 5253 |
| BatchNorm1d | $[-1, 51]$ | 102 |
| Dropout | $[-1, 51]$ | 0 |
| ReLU | $[-1, 51]$ | 0 |
| Linear (51, 1) | $[-1, 1]$ | 52 |

### 3.2. Convolutional Neural Network

We adopt a CNN [32] architecture in the encoder and decoder, which is the most basic and prominent model to learn the hierarchical representation of input data automatically in various domains like computer vision, image processing, and other tasks, which include grid-like structure data. The basic architecture of the CNN used in this paper is described in the Encoder and Decoder tables of Table 1, and is comprised as follows: (1) Convolutional 1D layer, the fundamental component of a CNN is the convolutional layer as it processes the input data to extract spectral features by performing a convolutional operation using filters/kernels, computing dot products among the local input and filter weights, and this layer outputs the set of features map. (2) Activation function, a most generally used, Rectified Linear Unit (ReLU) activation function is implemented on the output of the convolutional layer to adjust the negative values to zero, and ReLU instigates the non-linearity, which allows the model to understand the complex relationships among the features in the data. (3) Pooling layer, which is typically added after the convolutional layer to perform down-sampling by minimizing the spatial dimension of features while keeping the essential elements. Pooling captures the prominent features, lowers the computational cost, enhances the translation invariance, and the max pooling operation is usually applied where the max value in pooling is kept while the remaining values are removed. (4) Batch Normalization is commonly used to improve the model's training process and overall performance. It is applied after the convolution layers of the encoder. It normalizes the feature maps of each mini-batch by subtracting the mini-batch mean and dividing by the mini-batch standard deviation. This process helps reduce the internal covariate shift and makes the network more stable during training. (5) The Interpolate Layer, also known as an Upsampling layer or Deconvolution layer, is used to increase the spatial dimensions of an input feature map in the Decoder. It is the counterpart of the pooling or downsampling operation used in the Encoder. The Interpolate layer is used to resize the feature map to arbitrary dimensions. We used the nearest neighbor interpolation method, where each pixel in the output feature map is assigned the value of its nearest neighbor in the input feature map.

### 3.3. Multi-Layer Perceptron

The MLP is a type of ANN consisting of various fully connected layers called Linear. In Table 1, the subtable titled "MLP for regression" represents an MLP with two Linear layers, in this paper represented by [102, 51, 1], which means an input fully connected layer Linear (102,51) followed by an output fully connected layer Linear (51,1). In this study, we used an MLP only for regression tasks by exploiting the features extracted from the Encoder network. To attach the MLP to the Encoder output, we must flatten the output tensor from the dimension $(6 \times 17)$ to a single dimension $(102)$. A Dropout layer after each Linear layer, is used as a regularization technique. It is applied during training to prevent overfitting and improve the model's generalization capability. The MSE loss function for the MLP is the same as presented in Equation (1). The final regressor can be mathematically denoted as $R(E(X)) = P$, where $R(X)$ is the MLP function and $P$ represents the predicted CNC or CCC value. During the first training stage, the CNN-based autoencoder reduces

the reconstruction error between the original $X$ and reconstructed input $X'$. After that, the encoder-based latent space $E(X)$ provides dense features for HS signatures $X$. We used the SSL technique as a starting point in the training process, mainly the encoder's pre-trained weights as a base neural structure to attach the MLP model and predict the CNC and CCC values. For the experiments, we used different variants of the proposed model. (1) CNN-rand: our baseline model where all weights for $R(X)$ and $E(X)$ are randomly initialized and then modified during training; (2) CNN-static: where all weights for the pre-trained encoder $E(X)$ are frozen or kept static while the weights for the $R(X)$ function are learned during training; and (3) CNN-non-static: similar to the second variants, but the pre-trained weights for $E(X)$ function are fine-tuned.

## 4. Datasets

The following sections provide detailed information about the study area, field campaigns, and the data utilized in this paper.

### 4.1. Study Area and Field Campaigns

The study area is represented by two maize fields located near Grosseto (42°49′47.02″N 11°04′10.27″E; elev. 2 m a.s.l.), in Tuscany (Central Italy). According to the Köppen–Geiger classification, the area is categorized as Csa (Mediterranean climate), featuring a warm and temperate climate with much rainier winter months. The average annual temperature is 15.3 °C, and the yearly precipitations are about 749 mm. The two maize fields selected as test sites cover a total extension of more than 100 ha. The southern field was planted in early May, whereas the northern area was sown from the middle to the end of June, after the harvest of winter ryegrass. For this reason, the two fields showed different phenological conditions during the field campaigns and the aerial overpasses (Figure 3).

Two field campaigns were carried out from 2 to 7 July and from 31 July to 1 August 2018. following international protocols and guidelines [33,34], different crop traits, such as Leaf Area Index (LAI), leaf chlorophyll content (LCC), and leaf nitrogen content (LNC), were measured in several Elementary Sampling Units (ESUs), covering an area of $10 \times 10$ m$^2$. Indirect measurements of LAI and leaf chlorophyll content were acquired in 87 ESUs. Additionally, two sets of leaf discs for laboratory chlorophyll and nitrogen content extraction were collected by sampling the last fully-developed leaves of three plants in each ESU for a subset of 31 ESUs.

Indirect LAI measurements were carried out in all the 87 ESUs, using either the LAI2200 plant analyzer (LI-COR Inc., Lincoln, NE, USA) or digital hemispherical photography [34,35], according to the plant development stage. Indirect measurements of leaf chlorophyll content (LCC) were also acquired in every ESU using a SPAD-502 chlorophyll meter (Konica Minolta, Tokyo, Japan).

The first set of leaf discs was analyzed in laboratory analysis to extract leaf chlorophyll content (LCC [$\mu\,g\,cm^{-2}$]), based on the ice-cold methanol method. Chlorophyll values from laboratory extraction and SPAD measurements collected in the corresponding 31 ESUs, were used to identify a SPAD-LCC relation ($R^2 = 0.93$). This equation was then applied to the SPAD dataset to compute the LCC in all 87 ESUs. The second set of leaf discs was used to estimate the leaf nitrogen concentration (N$_{mass}$ [%]) through dry combustion with a CN elemental analyzer (Flash EA 1112 NC-Soil, Thermo Fisher Scientific, Pittsburgh, PA, USA) and leaf mass per area (LMA [$g\,cm^{-2}$]). LNC was calculated from N$_{mass}$ and LMA according to the following equation:

$$LNC = \frac{N_{mass} \cdot LMA}{100} \qquad \left[g\,cm^{-2}\right] \tag{3}$$

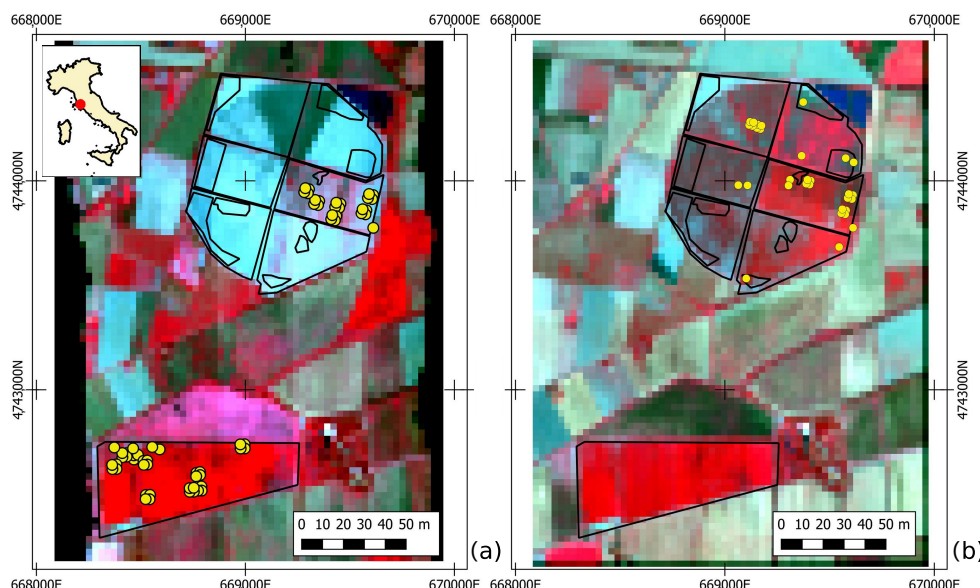

**Figure 3.** Study area, Elementary Sampling Units (ESU) location, and EO dataset acquired by HyPlant-DUAL pre-processed in CHIME-like configuration. Panel (**a**,**b**) reports 7 and 30 July images RGB (B43 820-B27 660-B15 540), respectively. Ground data collected in the two field campaigns are reported as yellow circles.

Finally, Canopy Chlorophyll Content (CCC) and Canopy Nitrogen Content (CNC) were calculated by multiplying LCC and LNC by the corresponding LAI:

$$CCC = \frac{1}{100} \cdot LCC \cdot LAI \qquad \left[ \text{g m}^{-2} \right] \tag{4}$$

$$CNC = 10000 \cdot LNC \cdot LAI \qquad \left[ \text{g m}^{-2} \right] \tag{5}$$

For further field measurements and laboratory analysis details, the reader can refer to [30].

As a result of this field campaign, we obtained two ground-truth datasets: the "*Grosseto CCC*" containing 87 measured samplings and the "*Grosseto CNC*" containing 31 measured samplings, representing CCC and CNC crop traits, respectively. The corresponding reflectance data for these two ground-truth datasets were extracted from the images described in Section 4.2, Earth Observation Dataset.

### 4.2. Earth Observation Dataset

The Earth Observation (EO) dataset includes two hyperspectral images acquired by the HyPlant-DUAL instrument [36–38] in the context of the project CHIME-RCS, founded by the European Space Agency (Figure 3). HyPlant is an airborne imaging spectrometer developed by Jülich Forschungszentrum in cooperation with SPECIM Spectral Imaging Ltd (Oulu, Finland). The spectrometer consists of two hyperspectral modules operating in a push-broom modality, namely measure reflectance (DUAL) and sun-induced chlorophyll fluorescence. The DUAL sensor measures contiguous spectral bands, from 370 to 2500 nm, with 3–10 nm spectral resolution in the VIS/NIR spectral range and 10 nm spectral resolution in the SWIR spectral range. The technical features of the acquired images are summarized in Table 2. The study area was acquired on 7 and 30 July with a ground sampling distance (GSD) of 1 m and 4.5 m, respectively. HyPlant-DUAL images were provided georectified and atmospherically corrected to top-of-canopy reflectance [39].

**Table 2.** Technical features of HyPlant-DUAL acquisitions over the study area, during the 2018 ESA FLEXSENSE campaign held in Grosseto.

| Date | Lines | Tot. Length | Tot. Area | Swath | GSD |
|------|-------|-------------|-----------|-------|-----|
| 7 July 2018 | 6 | ~7 km | ~18 km$^2$ | 400 m | 1 m |
| 30 July 2018 | 4 | ~8 km | ~20 km$^2$ | 1800 m | 4.5 m |

Both images were spatially and spectrally resampled to the expectation of the future CHIME mission to provide realistic CHIME reflectance data and maps of estimated crop traits. Spatial resampling to 30 m GSD was performed through a cubic convolution algorithm. Spectral resampling was performed considering theoretical Gaussian spectral response functions (i.e., 210 bands with 10 nm bandwidth). The bands influenced by atmospheric water vapor absorption were removed, leading to a final CHIME-like spectral configuration of 143 bands.

*4.3. Simulated Reflectances and Crop Traits Dataset*

This study estimated crop traits following a hybrid approach, which exploits an RTM to simulate the crop canopy reflectance, considering input parameters such as leaf and canopy variables and background, illumination, and viewing conditions (sun–target–sensor geometry). The RTM is run in forward mode to generate a database, or Look-Up-Table (LUT), which includes input-output pairs corresponding to the simulated reflectance spectra and the corresponding crop traits of interest (i.e., CCC and CNC). The generated LUT is then used to train DL algorithms to define a predictive relation between reflectance spectra and crop traits.

The RTM used in this study is the latest release of the PROSAIL model, which combines the leaf-level reflectance model PROSPECT-PRO [15] with the canopy-level reflectance model 4SAIL [40,41]. The LUT generation represents a critical step, as it should represent actual vegetation reflectance spectra [42]. To avoid unrealistic combinations of input variables in the PROSAIL-PRO model, a MATLAB script was designed to exploit (i) Probability Density Functions (PDFs), such as Uniform and Gaussian distributions, based on actual values observed during Grosseto 2018 field campaigns, and (ii) their covariances [30]. The full PROSAIL-PRO parameterization is summarized in Table 3. The final LUT includes 2000 reflectance spectra (each spectrum includes 143 reflectance values) and the corresponding crop traits. Further details regarding the LUT generation can be found in [30].

**Table 3.** List of PROSAIL-PRO input variables used to generate the LUT. Input variables were randomly sampled according to the reported distributions and ranges.

| | Param. | Description | Unit | PDF | Range [1] | |
|---|--------|-------------|------|-----|-------|---|
| PROSPECT-PRO | N | Structural parameter | - | Normal | 1.4 | 0.14 |
| | Cab | Chlorophyll content [2] | μg cm$^{-2}$ | Normal | 41.5 | 8.8 |
| | Ccx | Carotenoid content [2] | μg cm$^{-2}$ | Normal | 7.32 | 1.5 |
| | Canth | Anthocyanin content | μg cm$^{-2}$ | Normal | 0.0 | 0.0 |
| | Cbp | Brown pigment content | μg cm$^{-2}$ | Normal | 0.0 | 0.0 |
| | Cw | Water content [2] | mg cm$^{-2}$ | Normal | 12.92 | 1.91 |
| | Cp | Protein content [2] | g cm$^{-2}$ | Uniform | 0.0 | 0.001 |
| | CBC | Carbon-Based Constituents | g cm$^{-2}$ | Uniform | 0.003 | 0.006 |
| 4SAIL | ALA | Average Leaf Angle [2] | ° | Normal | 49.0 | 4.9 |
| | LAI | Leaf Area Index [2] | m$^2$ m$^{-2}$ | Normal | 1.77 | 1.4 |
| | HOT | Hot spot parameter | m m$^{-1}$ | Normal | 0.01 | 0.001 |
| | SZA | Solar Zenith Angle [2] | ° | Uniform | 26 | 30 |
| | OZA | Observer Zenith Angle | ° | Uniform | 0 | 0 |
| | RAA | Relative Azimuth Angle | ° | Uniform | 0 | 0 |
| | BG | Soil Spectra [2] | - | Uniform | 2 | 4 |

[1] min and max values in case of Uniform PDF; $\mu$ and $\sigma$ values in case of Normal PDF. [2] Ranges set according to values measured in this study.

## 5. Experiments, Results and Discussion

To understand the potentialities and the weakness of the proposed solution, we conducted two main groups of experiments:

- In the first group, to evaluate the self-supervised approach, we fixed the autoencoder topology and compared a large set of different MLP configurations combined with different encoder initializations. An ablation study was also conducted;
- In the second group of experiments, to assess the performance of the proposed solution with a standard features extraction approach, the accuracies of MLP estimations using features extracted from the Encoder were compared with those extracted by a Principal Component Analysis (PCA) with different configurations;
- Finally, a comparison with a previously published result is also presented.

### 5.1. Metrics

To evaluate the proposed SSL model, we used two different measures, namely the $R2$ and the RMSE.

The $R2$ metric, also known as the coefficient of determination, is a commonly used statistical measure to evaluate the performance of a regression model. It indicates how well the model fits the observed data. $R2$ is a value between 0 and 1, where 0 indicates that the model does not explain any of the variability in the data, and 1 indicates that the model perfectly explains all the variability. Still, it can be negative (because the model can be arbitrarily worse). To calculate $R2$, first, the Total Sum of Squares ($TSS$) is computed, representing the total variability in the dependent variable. Then, the sum of squares of residuals ($RSS$) is calculated, representing the unexplained variability or the model's error. Finally, the $R2$ score is obtained by subtracting the ratio of $RSS$ to $TSS$ from 1:

$$R2 = 1 - (RSS/TSS) \tag{6}$$

The RMSE (Root Mean Square Error) is another popular metric used to evaluate the performance of a regression model. It measures the average magnitude of the residuals or prediction errors of the model. The RMSE is particularly useful because it captures the errors' direction and magnitude. Mathematically, the RMSE is computed as follows:

$$RMSE = \sqrt{(1/n) \sum (y_{pred} - y_{actual})^2} \tag{7}$$

where $n$ is the number of data points, $y_{pred}$ are the predicted values from the regression model, and $y_{actual}$ are the actual values of the dependent variable.

### 5.2. Autoencoder Model

In the first step, we train a convolutional autoencoder using the Earth Observation (EO) dataset to de-noise the hyperspectral signatures presented as input to the network. All pixels of the image acquired on 7 July 2018 were used for training, while the image acquired on 30 July 2018 was used for validation. Therefore, the training set for the autoencoder model contains 5146 samples, while the validation set contains 5292 samples. As described in Section 3, we used data augmentation to increase the variability inside the dataset. When we add noise to the input signal, we do not need supervision to reconstruct the original HS signature in the output. We use the original HS signature plus augmentation as ground truth. The autoencoder was trained for 1000 epochs using the SGD optimizer with a learning rate equal to 0.001. The plot in Figure 4 reports the training and validation loss values for all the training epochs. We can see how the model is not overfitting and continues reducing the loss value on the training and validation images. In Figure 5, we can analyze the predictions over a training and validation sample. The predicted HS signal is similar to the target, and the model can remove the random input noise very well.

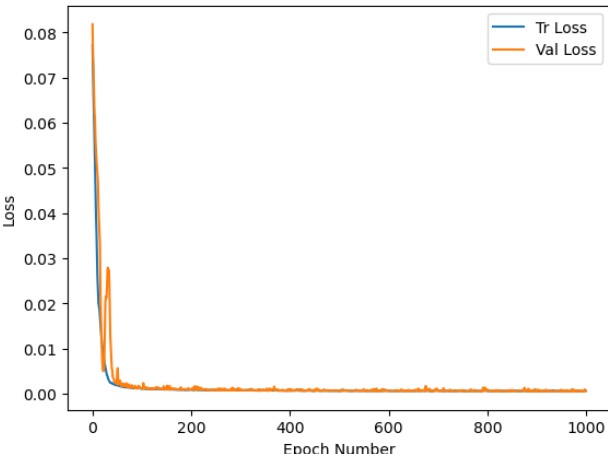

**Figure 4.** Training and validation loss for the autoencoder trained for 1000 epochs with unsupervised hyperspectral data.

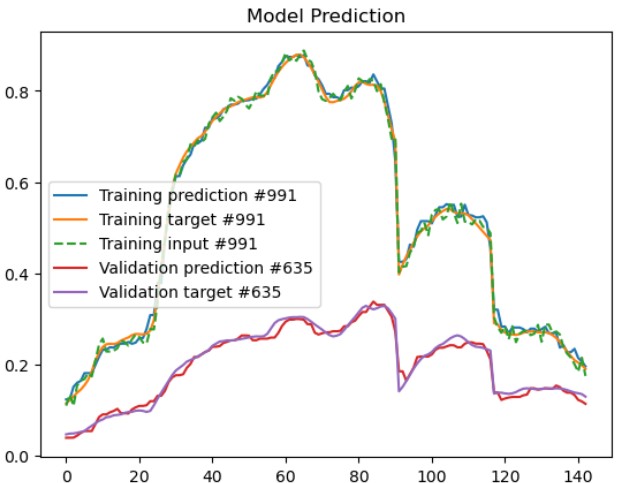

**Figure 5.** Autoencoder training and validation input/target/prediction examples.

### 5.3. Model for Regression

In this group of experiments, the same autoencoder topology was used, and its details are reported in Table 1. To evaluate the benefit introduced by the two phases of the proposed SSL learning method, we changed many configurations for the MLP added on top of the Encoder. Since real samples measured in the field are very limited in number, we used the Simulated Reflectances and Crop Traits Dataset presented in Section 4.3 for training and validation of the regressor. In this way, the training set for the regressor model contains 2000 simulated reflectances and crop traits' samples, whereas the test set contains 87 and 31 real reflectances and crop traits' samples, measured in the field, for CCC and CNC, respectively. The first two experiments compare the benefits introduced by fine-tuning with those of using the static weights of the pre-trained Encoder. Finally, using an ablation study, we try to understand the advantages of SSL. In Figure 6, we reported a graphical representation of all the $R2$ and RMSE measures from these experiments. All the numerical results are reported in the Tables of the Supplementary Materials. As noted from the same figure, we varied the learning rate in the set $\{0.05, 0.01, 0.005\}$ and the MLP topology for the final Encoder connected to the MLP model. In particular, we tested many different configurations of the MLP in terms of the number of layers and number of neurons per layer. The "MLP Topology" axes in Figure 6 show two configuration groups: one with three and another with four layers (the number of neurons per MLP's layer is reported inside the square brackets). For the estimation of the CNC parameter, we obtained the best absolute ($R2 = 0.9186$ and $RMSE = 0.7908$ using 0.01

for learning rate, [102, 51, 1] for MLP topology and Non-static for the Encoder initialization) and average measures when we fine-tuned the pre-trained Encoder (Non-static column). Instead, we obtained the best absolute ($R2 = 0.8318$ and $RMSE = 0.2490$, using 0.01 for learning rate, [102, 102, 102, 1] for MLP topology and Static for the Encoder initialization) and average measures using the frozen pre-trained Encoder (Static column) to estimate the CCC parameter. Thanks to the color scales, we can see that the best solution is to use a static or fine-tuned configuration for the Encoder parameters. This means that the features learned by the autoencoder are the best and help to improve the results. No particular trend emerges on the MLP's topology or learning rate value to use. As a final test, we computed the crop trait maps for CCC and CNC using the best models obtained to estimate CCC and CNC parameters. As shown in Figure 7, and confirmed by the comparison with ground-truth data of CCC and CNC, the model can generalize and predict the values of CCC and CNC on all images with reasonable precision. For the ablation study, analyzing the last two columns (Static and Non-static) of Figure 6 for both CCC and CNC parameters, the average and absolute measurements are always much better than the measurements obtained with the same configurations but randomly initializing the Encoder weights. This demonstrates that the self-supervised learning methodology still brings advantages to the final resulting model for HS data.

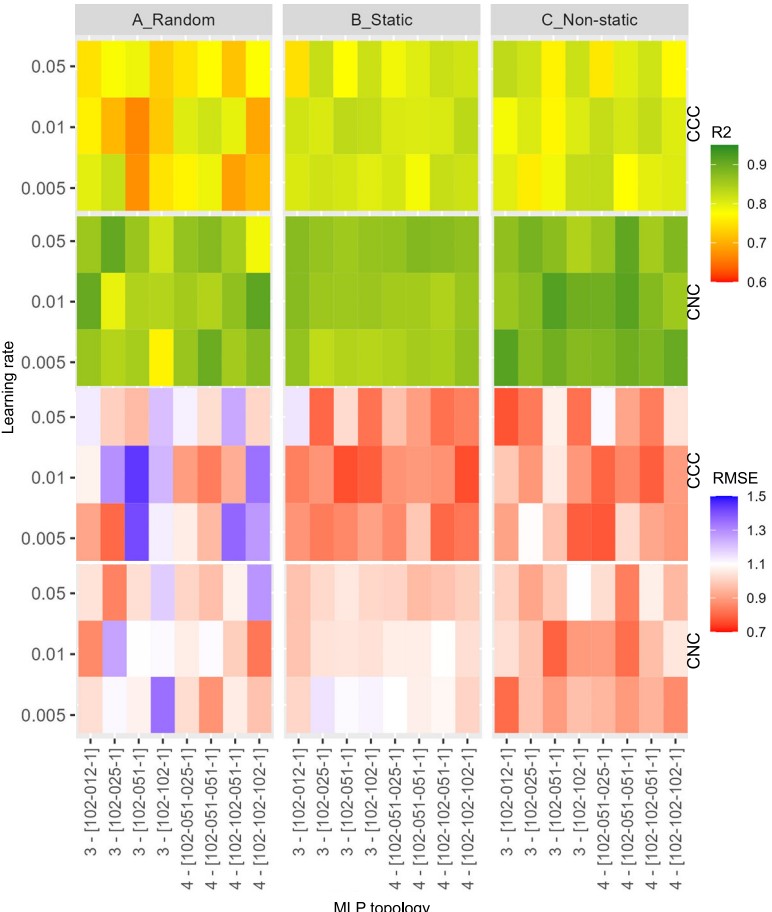

**Figure 6.** $R2$ (the two top lines) and RMSE (the two bottom lines) results obtained on Grosseto CCC and CNC test sets using different hyperparameters to estimate the CNC and CCC parameters. The model was trained for 1500 epochs for each experiment using the SGD optimizer. The Random, Static, and Non-static columns are the three ways the Encoder weights have been used. The best results for CCC are $R2 = 0.8318$ and $RMSE = 0.2490$ (using 0.01 for learning rate, [102, 102, 102, 1] for MLP topology and Static for the Encoder), while for CNC are $R2 = 0.9186$ and $RMSE = 0.7908$ (using 0.01 for learning rate, [102, 51, 1] for MLP topology and Non-static for the Encoder).

Based on the results shown above, the scatter plots of the best models based on SSL are reported in Figure 8, where the position of the points in the plot shows how the outputs of the proposed model are pretty faithful to the measured values.

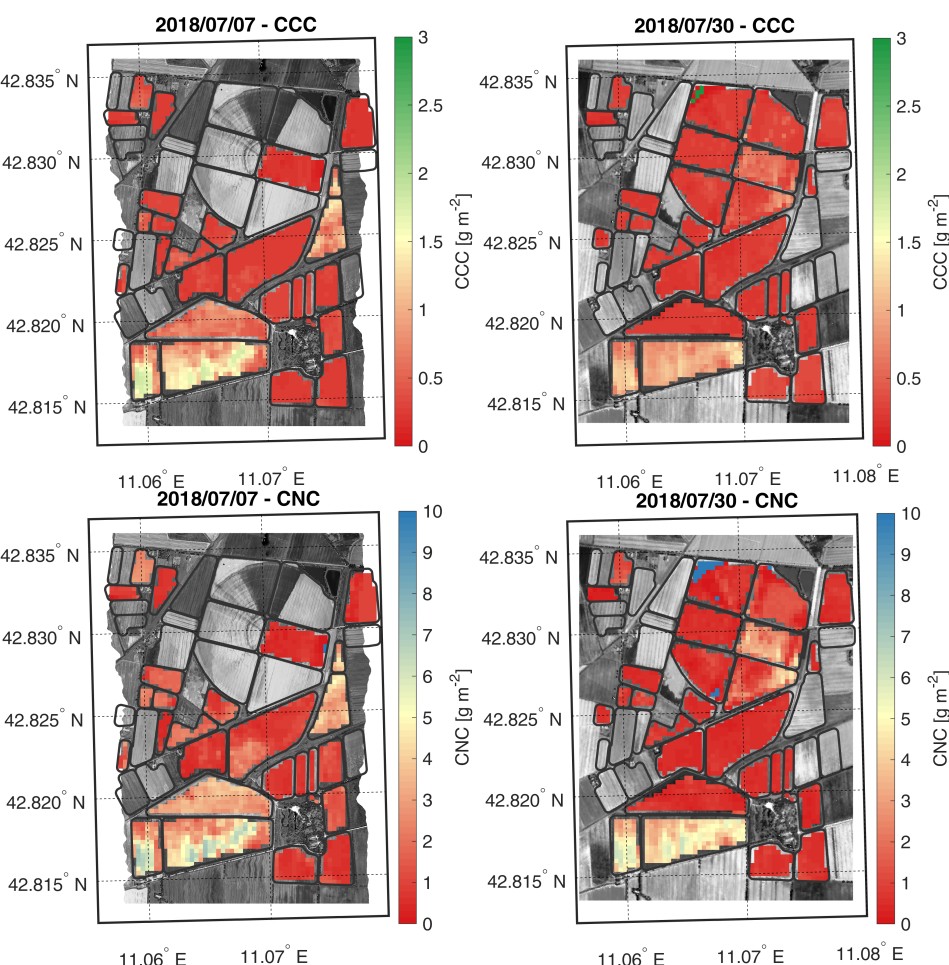

**Figure 7.** Regression maps of CCC (**top row**) and CNC (**bottom row**) generated from the best model obtained, starting from CHIME-like synthetic images acquired on July 7 and 30, 2018, belonging to the Earth observation dataset.

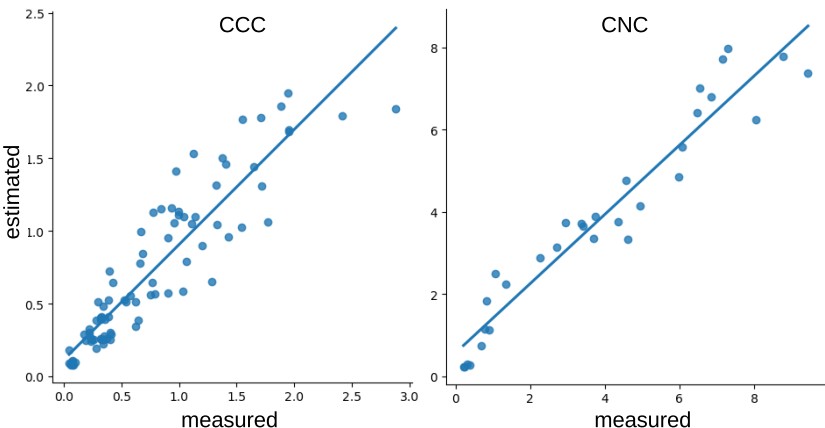

**Figure 8.** Scatter plots for the best results reported in Figure 6.

*5.4. Comparisons*

The data set used in this study was previously exploited in two other works. In [43], the authors evaluated the retrieval of CCC and CNC from synthetic PRISMA data. Among several tested ML algorithms (Gaussian Process Regression (GPR), Artificial Neural Networks (NN), Partial Least Square Regression (PLSR), Random Forests (RF), and Support Vector Regression (SVR)), the GPR model achieved the best accuracy for both CCC and CNC. A further step was performed by [30], where the GPR algorithm was applied to the same data set to assess chlorophyll and nitrogen content, at both leaf and canopy level. The best results achieved in this study were obtained using a hybrid framework, including an active learning (AL) technique for the optimization of the sample size of the training database. Considering the above findings, this section compares the results of the deep learning solution proposed in this study (Encoder + MLP) with (i) those achieved using the classical PCA feature extraction techniques (PCA + MLP), and (ii) the results presented in [30] (PCA + GPR, PCA + GPR-AL).

To compare our SSL solution with a classical approach typically used in the field of HS images, we used the statistical Principal Component Analysis (PCA) technique to reduce the dimensionality of a dataset and preserve the maximum amount of information. We used four different values for the number of components to keep; in particular, we trained PCA on LUT presented in Section 4.3 to extract 5, 10, 15, and 20 components. Starting from the PCA output, we used the same MLP configurations used in the previous experiment to compare the regressor with a different technique than the one proposed. The best results are shown in Table 4, which are never higher than those obtained with the proposed solution. Therefore, this experiment demonstrates that the proposed solution introduces advantages compared to classic dimensionality reduction techniques usually adopted in machine learning solutions. All the numerical and graphical results are reported in the Supplementary Materials. We can observe how the best results obtained with PCA are always lower (CCC $R2$ 0.82, RMSE 0.26; CNC $R2$ 0.88, RMSE 0.93) than those obtained using the SSL technique (CCC $R2$ 0.83, RMSE 0.25; CNC $R2$ 0.92, RMSE 0.79) in particular for CNC.

For the second comparison, we reported in Table 4 the results presented in [30]. This paper applies two solutions to the PCA features extracted from the same LUT exploited in this study. The first solution uses the GPR model on PCA simulated data; the best results are lower than those obtained with the proposed model (CCC $R2$ 0.79, RMSE 0.38; CNC $R2$ 0.84, RMSE 1.10). The second solution uses an AL technique (PCA + GPT* in Table 4), which uses the Grosseto CCC and CNC test sets in the training process to optimize the simulated LUT (smart sampling) dynamically. This solution shows the lowest RMSE for both CCC (0.21) and CNC (0.71), but they are obtained in cross-validation, not on an independent dataset.

**Table 4.** Comparison between the results obtained on Grosseto CCC and CNC data sets. The proposed deep learning solution (Encoder+MLP) was compared with (i) PCA for feature extraction (FE) and MLP as ML model for regression, and (ii) the results presented in [30] that use PCA for FE and either GPR or GPR with active learning (GPR-AL) as ML model.

| FE | ML | VAR | *R2* | RMSE |
|----|-----|-----|------|------|
| Encoder | MLP | CCC | 0.8318 | 0.2490 |
| PCA | MLP | CCC | 0.8275 | 0.2521 |
| PCA | GPR | CCC | 0.79 | 0.38 |
| PCA | GPR-AL | CCC | 0.88 | 0.21 |
| Encoder | MLP | CNC | 0.9186 | 0.7908 |
| PCA | MLP | CNC | 0.8861 | 0.9351 |
| PCA | GPR | CNC | 0.84 | 1.10 |
| PCA | GPR-AL | CNC | 0.93 | 0.71 |

## 6. Conclusions

In conclusion, this paper introduces an innovative two-stage learning method for retrieving crop traits, specifically chlorophyll canopy content (CCC) and nitrogen canopy content (CNC), using hyperspectral imagery acquired during the CHIME-RCS project founded by the European Space Agency. The proposed method leverages self-supervised learning (SSL) techniques, which enable the extraction of valuable representations from unlabeled data without needing human-annotated labels. The two-stage learning approach consists of pre-training a de-noising Convolutional Neural Network (CNN) with pairs of corrupted and clean images, followed by fine-tuning the network using SSL strategies to capture spectral correlations.

The results presented in the paper demonstrate the effectiveness of the proposed technique in estimating the Cab and N content of maize crops from hyperspectral images. The method achieves excellent prediction results of crop traits by leveraging the information content of simulated space-borne HS data such as the one provided by the CHIME mission. This approach holds promise for advancing remote sensing techniques in agriculture, allowing for non-invasive and accurate monitoring of crop traits at the canopy level.

The findings of this study contribute to the field of agricultural remote sensing by showcasing the potential of SSL methods in hyperspectral data analysis. By eliminating the dependency on human-annotated labels, this approach offers a cost-effective and scalable solution for extracting valuable information from large-scale unlabeled datasets. The proposed two-stage learning method can serve as a foundation for further research and development of advanced techniques in crop trait estimation using hyperspectral imaging data already available from the PRISMA-ASI and ENMAP-DLR missions, opening new opportunities to enhance our understanding of plant health and supporting precision farming activities and sustainable agriculture practices.

**Supplementary Materials:** The following supporting information can be downloaded at: https://www.mdpi.com/article/10.3390/rs15194765/s1, Figure S1: Graphical representation of the CCC estimation using MLP and PCA; Figure S2: Graphical representation of the CNC estimation using MLP and PCA; Table S1: CNC estimation using Self-Supervised learning; Table S2: CCC estimation using Self-Supervised learning; Table S3: CCC estimation using MLP and PCA; Table S4: CNC estimation using MLP and PCA.

**Author Contributions:** Conceptualization, I.G.; Methodology, I.G.; Software, I.G.; Validation, M.B. and G.C.; Investigation, I.G.; Data curation, M.B.; Writing—review & editing, A.U.R. and G.C.; Supervision, M.B. and G.C. All authors have read and agreed to the published version of the manuscript.

**Funding:** The research activities have been framed in the CNR DIPARTIMENTO DI INGEGNERIA, ICT E TECNOLOGIE PER L'ENERGIA E I TRASPORTI project "DIT.AD022.207/STRIVE (FOE 2022)", sub-task activity "Agro-Sensing". Methodological development is a contribution to the "PRR.AP002.005 SPOKE3 Enabling technologies and sustainable strategies for the smart management of agricultural systems and their environmental impact (AGRITECH)".

**Data Availability Statement:** The data that support the findings of this study are available from the corresponding author, G.C., upon reasonable request.

**Conflicts of Interest:** The authors declare no conflict of interest.

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
