# Peer review of "Self-Supervised Convolutional Neural Network Learning in a Hybrid Approach Framework to Estimate Chlorophyll and Nitrogen Content of Maize from Hyperspectral Images"

_remotesensing, doi:10.3390/rs15194765_

Round 1
Reviewer 1 Report
This manuscript proposes a CNN-based hybrid method to estimate Chlorophyll and Nitrogen Content of Maize from hyperspectral imagery. Overall, the topic is important and interesting, and the experiment validated the effectiveness of the method. However, there are some issues should be clarified.
1. In the abstract part, SSL technique could be introduced more briefly.
2. How many training samples have you used in your experiment?
3. The content of part 5. Metrics is relatively less, can it be combined with other sections?
4. Is it convenient to compare your method with the state-of-the-art methods?
5. Please give the references for the content in lines 70-71.
Author Response
This manuscript proposes a CNN-based hybrid method to estimate Chlorophyll
and Nitrogen Content of Maize from hyperspectral imagery. Overall, the topic
is important and interesting, and the experiment validated the effectiveness of
the method. However, there are some issues should be clarified.
-
In the abstract part, SSL technique could be introduced more briefly.
We have replaced the previous explanation of the SSL technique in the abstract section with a more concise introduction (text coloured in blue).
-
How many training samples have you used in your experiment?
In each subsection of section "6. Experiments, Results and Discussion", we reported (in blue) the number of training samples used to train all the models.
-
The content of part 5. Metrics is relatively less, can it be combined with other sections?
"Metrics" has been moved inside the section "Experiments, Results and Discussion" section as a subsection.
-
Is it convenient to compare your method with the state-of-the-art methods?
To our knowledge, the state-of-the-art method to assess crop traits is the hybrid approach, which couples radiative transfer models (RTMs) and artificial intelligence techniques. In recent years, several papers have tested different algorithms within a hybrid framework for retrieving crop traits. In particular, the other two previous works tested the hybrid approach on the same data set used in this study. In Candiani et al., 2022 [31], the GPR algorithm was used to assess chlorophyll and nitrogen content at leaf and canopy levels through a hybrid approach. This work also tested active learning (AL) heuristics to optimize the sample size of the database used for the training, which provided better results than the classic hybrid approach. In Ranghetti et al., 2022 [44], several ML algorithms (Gaussian Process Regression (GPR), Artificial Neural Networks (NN), Partial Least Square Regression (PLSR), Random Forests (RF) and Support Vector Regression (SVR)) were tested following the classic hybrid approach, to assess CCC and CNC. The comparison between the different ML algorithms showed that the GPR model achieved the best accuracy for both parameters, confirming the results obtained by Candiani et al., 2022 [31].
The following text was added to the manuscript:
“The data set used in this study was previously exploited in two other works. In [44], the authors evaluated the retrieval of CCC and CNC from synthetic PRISMA data. Among several tested ML algorithms (Gaussian Process Regression (GPR), Artificial Neural Networks (NN), Partial Least Square Regression (PLSR), Random Forests (RF) and Support Vector Regression (SVR)), the GPR model achieved the best accuracy, for both CCC and CNC. A further step was performed by [31], where the GPR algorithm was applied to the same data set to assess chlorophyll and nitrogen content at both leaf and canopy levels. The best results achieved in this study were obtained using a hybrid framework, including an active learning (AL) technique for the optimization of the sample size of the training database. Considering the above findings, this section compares the results of the deep learning solution proposed in this study (Encoder+MLP) with i) those achieved using the classical PCA feature extraction techniques (PCA+MLP) and ii) the results presented in [31] (PCA+GPR, PCA+GPR-AL).”
-
Please give the references for the content in lines 70-71.
We have changed the sentence slightly to make the sentence better understood.
Reviewer 2 Report
This study focuses on Self-supervised CNN Learning in a Hybrid Approach Framework to Estimate Chlorophyll and Nitrogen Content of Maize from Hyperspectral Images. The topic of this work is to Estimate Chlorophyll and Nitrogen Content of Maize from Hyperspectral Images, therefore, falls within the scope of the journal. This paper is very nicely written, all the sections (methods, results, and discussion, and conlcusion) are adequately described. The paper may be aacepted in its present form. I appreciate authors for doing very good research work and in recent context, these type of studies are the need of hour.
I have only one suggestion that if authors have tried other algorithm than CNN, beacuse in this case accuracy is important or they can quote some literatures in which such type of work has been done.
Author Response
-
I have only one suggestion that if authors have tried other algorithm than CNN, beacuse in this case accuracy is important or they can quote some literatures in which such type of work has been done.
On the same dataset used in this study, other ML regression algorithms were tested in two previous works. In Candiani et al., 2022 [31], the GPR algorithm was used to assess chlorophyll and nitrogen content at leaf and canopy levels through a hybrid approach. This work also tested active learning (AL) heuristics to optimize the sample size of the database used for the training, which provided better results than the classic hybrid approach. In Ranghetti et al., 2022 [44], several ML algorithms (Gaussian Process Regression (GPR), Artificial Neural Networks (NN), Partial Least Square Regression (PLSR), Random Forests (RF) and Support Vector Regression (SVR)) were tested to assess CCC and CNC. The comparison between the different ML algorithms showed that the GPR model achieved the best accuracy for both parameters, confirming the results obtained by Candiani et al., 2022 [31].
The following text was added to the manuscript:
“The data set used in this study was previously exploited in two other works. In [44], the authors evaluated the retrieval of CCC and CNC from synthetic PRISMA data. Among several tested ML algorithms (Gaussian Process Regression (GPR), Artificial Neural Networks (NN), Partial Least Square Regression (PLSR), Random Forests (RF) and Support Vector Regression (SVR)), the GPR model achieved the best accuracy, for both CCC and CNC. A further step was performed by [31], where the GPR algorithm was applied to the same data set to assess chlorophyll and nitrogen content at both leaf and canopy levels. The best results achieved in this study were obtained using a hybrid framework, including an active learning (AL) technique for the optimization of the sample size of the training database. Considering the above findings, this section compares the results of the deep learning solution proposed in this study (Encoder+MLP) with i) those achieved using the classical PCA feature extraction techniques (PCA+MLP) and ii) the results presented in [31] (PCA+GPR, PCA+GPR-AL).”
Reviewer 3 Report
Did you look for this correlation between the SPAD values and the individual hyperspectral reflectance values?
It may be interesting to compare the SPAD values with the NIR/RED ratio calculated from the same hyperspectral wavelength range.
Author Response
-
Did you look for this correlation between the SPAD values and the individual hyperspectral reflectance values? It may be interesting to compare the SPAD values with the NIR/RED ratio calculated from the same hyperspectral wavelength range.
We have performed this kind of evaluation with other datasets, comparing SPAD values with single bands, simple band ratio and/or 2-band normalized difference index. The relationships between the best bands/bands-combinations and SPAD are usually valid only on the dataset they were developed for, lacking transferability on other domains (space) or different vegetative seasons (time).
In recent years, the scientific community has moved from empirical methods to the so-called hybrid method, which couples radiative transfer models (RTMs) and artificial intelligence techniques to overcome this issue. RTMs, which implement the physically-based processes occurring in the plants, are independent of a measured dataset and should guarantee the transferability of proposed methods to other contexts. Since this paper proposed a hybrid method to assess chlorophyll and nitrogen content, the comparison of SPAD values with the NIR/RED ratio (or other spectral indices) was not performed, as considered in the scope of the paper.
Round 2
Reviewer 1 Report
The authors have answered my questions, I suggested it to be published.